# DNA Methylation Module Network-Based Prognosis and Molecular Typing of Cancer

**DOI:** 10.3390/genes10080571

**Published:** 2019-07-28

**Authors:** Ze-Jia Cui, Xiong-Hui Zhou, Hong-Yu Zhang

**Affiliations:** Hubei Key Laboratory of Agricultural Bioinformatics, College of Informatics, Huazhong Agricultural University, Wuhan 430070, China

**Keywords:** Cancer, DNA methylation, module network, prognostic analysis, molecular typing

## Abstract

Achieving cancer prognosis and molecular typing is critical for cancer treatment. Previous studies have identified some gene signatures for the prognosis and typing of cancer based on gene expression data. Some studies have shown that DNA methylation is associated with cancer development, progression, and metastasis. In addition, DNA methylation data are more stable than gene expression data in cancer prognosis. Therefore, in this work, we focused on DNA methylation data. Some prior researches have shown that gene modules are more reliable in cancer prognosis than are gene signatures and that gene modules are not isolated. However, few studies have considered cross-talk among the gene modules, which may allow some important gene modules for cancer to be overlooked. Therefore, we constructed a gene co-methylation network based on the DNA methylation data of cancer patients, and detected the gene modules in the co-methylation network. Then, by permutation testing, cross-talk between every two modules was identified; thus, the module network was generated. Next, the core gene modules in the module network of cancer were identified using the K-shell method, and these core gene modules were used as features to study the prognosis and molecular typing of cancer. Our method was applied in three types of cancer (breast invasive carcinoma, skin cutaneous melanoma, and uterine corpus endometrial carcinoma). Based on the core gene modules identified by the constructed DNA methylation module networks, we can distinguish not only the prognosis of cancer patients but also use them for molecular typing of cancer. These results indicated that our method has important application value for the diagnosis of cancer and may reveal potential carcinogenic mechanisms.

## 1. Introduction

The most important concern for cancer patients is to understand the likely course and chances of recovery, which is the prognosis of cancer. Many factors affect the prognosis of cancer [1]. The classification of cancer is one of the most prominent of these factors. As a complex polygenic disease, the same cancer performs differently in different individuals, and the same clinical manifestations may require different treatment options. The heterogeneity of cancer makes it impossible to assess tumors by relying on limited clinical indicators, which reflects the need to study cancer at the molecular level [2].

In recent years, the rapid development of high-throughput sequencing technology and microarray technology has enabled researchers to systematically study cancer at the molecular level. For example, molecular biomarkers that predict the prognosis of cancer patients are found based on gene expression data [3,4]. However, these prognostic genes often have poor generalization ability [5], and most of these genes are not oncogenes but noise signals [6]. In addition, with advances in epigenetics research, the importance of DNA methylation abnormalities in cancer has gradually emerged [7,8]. DNA methylation is an important epigenetic modification that does not alter the DNA sequence, which is essential for the development, progression, and metastasis of cancer [9,10]. Some DNA methylation biomarkers have been used to guide clinical practice [11]. The genomic coverage of the DNA methylation microarray platform is increased, and the cost is reduced, which has prompted us to study the pathogenesis of cancer at the methylation level.

Cellular function is performed by a module composed of a variety of interacting molecules [12]. In particular, cancer is a system of multigene expression patterns and functional modules that are constantly changing, and it seems that the gene modules outperform the gene signatures in prognosis and molecular typing. Modules are not isolated, and there is also cross-talk among them [13]. However, most studies have ignored the cross-talk [14,15], some important modules related to cancer might be overlooked.

Based on the above theories, in our research, we collected DNA methylation data, gene expression data, and corresponding clinical data (including survival time and survival status) of breast invasive carcinoma (BRCA), skin cutaneous melanoma (SKCM) and uterine corpus endometrial carcinoma (UCEC) with abundant samples in The Cancer Genome Atlas (TCGA). First, we evaluated the stability in cancer prognosis of DNA methylation data and gene expression data for each of the three cancers and proved that DNA methylation data are more suitable for cancer prognosis research. Then, DNA methylation data were used to construct gene co-methylation networks for the three cancers using the rank-based method and to identify gene modules in three co-methylation networks. Next, we used the method of permutation testing to calculate the cross-talk between every two modules, thereby forming a module network, and then we found the core gene modules in the module network by the K-shell method. Finally, these core gene modules are used as features to study the prognosis and molecular typing of cancer and are evaluated by survival analysis.

## 2. Materials and Methods

### 2.1. Data Collection and Preprocessing

The datasets of breast invasive carcinoma (BRCA), skin cutaneous melanoma (SKCM) and uterine corpus endometrial carcinoma (UCEC) were downloaded from The Cancer Genome Atlas (TCGA, https://tcga-data.ncbi.nih.gov/tcga/) [16]. In each dataset, it contained the DNA methylation data, mRNA expression data and clinical data (time of death and death status). There were 780 samples in BRCA, 468 samples in SKCM, and 428 samples in UCEC that matched DNA methylation data, mRNA expression data and clinical data. 

In this work, the mRNA expression data were measured by RNA-seq, and the probes of expression data were mapped to Gene Symbol. The FPKM (fragments per kilobase of transcript per million mapped reads) value for each gene was used to represent the expression level. For the DNA methylation data that was measured by the Illumina HumanMethylation 450K Assay, the methylation level of the gene was represented by the β value, and the following filtering criteria were applied: removal of probes that the β values are NA in the samples, removal of probes targeting the X and Y chromosomes, removal of all probes affected by SNPs [17], and filtering out of probes that have been shown to be cross-reactive [17]. Finally, the probes that fell in the promoter region were selected and mapped to Gene Symbol. The methylation levels of the probes for each gene were averaged, and after converting the β value of each gene into the M value, the transformation relationship was determined to be as follows [18]
(1)$$M=log_{2}\frac{\beta }{1-\beta }$$

The detailed results of preprocessing are shown in Appendix A.

### 2.2. Comparison of the Stability of DNA Methylation Data and Gene Expression Data in Cancer Prognosis

In the prognosis of cancer, the largest problem is the stability of the prognostic genes that are identified based on high-throughput data. We used a method to examine the stability of gene expression data and DNA methylation data for three cancers in prognosis [19]. First, gene expression data (or DNA methylation data) for each cancer were randomly divided into two groups of equal numbers. Then, Cox regression was applied to select the genes of each group whose expression levels (or methylation levels) were significantly associated with the prognosis of cancer patients (*p*-value < 0.05). After that step, the hypergeometric distribution test was used to evaluate whether the overlap of the two prognostic gene sets obtained from the two groups was significant. By repeating the above steps 100 times, we can obtain 100 *p*-values of the hypergeometric distribution test for each of the two data types. We defined the negative logarithm of *p*-values as an indicator of the stability of the dataset. Finally, the Mann–Whitney–Wilcoxon test was used to verify whether there is a significant difference in the stabilities of the gene expression datasets and the DNA methylation datasets [19].

### 2.3. Construction of the Gene Co-Methylation Networks by Rank-Based Method

In general, participation in a common pathway or functional similarity leads to gene coexpression [20,21]. This property can also be applied to co-methylation [22], and some studies have adopted a few methods to construct the co-methylation network [19,22]. The common methods for constructing gene coexpression networks can be divided into two categories: one is the value-based method (utilize the similarity values) [23,24], and the other is the rank-based method (utilize the rank-transformed similarities) [25,26]. The value-based method is significantly limited by the homogeneous threshold for all genes in the network [27]. In fact, genes in different functional pathways are regulated by different mechanisms, some genes in one pathway may be strongly mutually coexpressed, while some genes in another pathway may be weakly coexpressed [27]. Therefore, it may be more reasonable to construct the gene coexpression network based on the rank-based method. According to the above theory, we applied the rank-based method to construct the co-methylation network. First, we used the Pearson correlation coefficient to calculate the correlations of the methylation levels between every two genes. Then, based on the Pearson correlation coefficient, for each gene, we selected only the 4 most relevant sites as its neighbors [27]; thus, all selected pairs of DNA methylation genes constituted a co-methylation network.

### 2.4. Gene Module Detection in Co-Methylation Networks

Cytoscape 3.6.1 was used to visualize the gene co-methylation networks of the three cancers [28]. In addition, the MCODE plug-in for Cytoscape was used to detect the gene dense clusters in the network [29], and only the modules that contained no fewer than five genes were retained.

### 2.5. Construction of the Module Networks by Permutation Method

In the co-methylation network, if the number of edges between two modules is significantly higher than random, there may be cross-talk between the two modules. The significance of the cross-talk between every two modules was calculated by permutation test [19]. We proceeded as follows. First, the number of edges across the two modules in the co-methylation network was calculated. Second, we selected two random gene sets that contain the same number of genes as the two real modules in the co-methylation network and calculated the number of edges across the random gene sets. Then, step 2 was repeated 1000 times, the number of edges across the random gene sets was set as the null hypothesis distribution, and the p-value of the cross-talk between the two modules was calculated based on the null hypothesis distribution. According to the permutation test, all significant pairs (*p*-value < 0.05) across modules could construct a gene module network.

### 2.6. Identifying Core Gene Modules Based on the K-Shell Method

Based on the gene module network, we adopted the K-shell method to identify the core gene modules. The advantage of the K-shell method is that it considers the relative location of the node in the network [30]. Deciphering the network architecture by this method could help to discover novel components in complex systems [31]. In this study, each node represents a gene module, and we finally found the core gene modules in the gene module networks of the three cancers using the K-shell method. Then, these core gene modules were used as features for subsequent analysis.

### 2.7. Survival Analysis Using Core Gene Modules

After obtaining the core gene modules, we applied a strategy similar to the GGI (gene expression grade index) to calculate the prognostic risk of each patient [32]:(2)$$\mathrm{Prognosis}\;\mathrm{Risk}=\;\sum x_{i}-\sum y_{j}$$

For every cancer, we randomly divided samples into equal numbers of training datasets and test datasets. The prognostic risk for each patient in the test set was calculated by Formula 2. In this equation, $x_{i}$ is the statistical value (the average value of all genes’ methylation levels in the module) of the module with a positive Cox coefficient, and $y_{i}$ is the statistical value of the module with a negative Cox coefficient [33]. Then, the samples in the dataset were divided into two groups with the same number of samples based on their prognostic risks. In the end, we used the log rank test to test whether there was a significant difference in the patients’ overall survival between the two groups.

### 2.8. Cancer Molecular Typing Based on Core Gene Modules by K-Means Algorithm

We used the K-means algorithm to perform molecular typing on cancer patients. Normalization of data is required before the K-means, and we used the min-max standardization method. To determine the size of cluster number k, we applied the silhouette coefficient to judge the clustering quality in different k values (k ∈ [2,8]) [34]. The calculation method is as follows:(3)$$S\left(i\right)=\;\frac{b\left(i\right)-a\left(i\right)}{\max\left\{a\left(i\right),b\left(i\right)\right\}}$$

The K-means method divides the data into k clusters. For each point i in the cluster, a(i) represents the average of the distance of the i vector to all other points in the cluster to which it belongs; b(i) represents the minimum of the average distance of the i vector to all points of each cluster to which it does not belong. Finally, the K-means clustering quality at this k value is the silhouette coefficients of all points at average. 

## 3. Results and Discussion

### 3.1. DNA Methylation Data are More Stable in Cancer Prognosis

We systematically evaluated the stability of gene expression data and DNA methylation data for three cancers by the overlap of the prognostic genes selected from different samples. The detailed definition of data stability is presented in Section 2.2, and the evaluation results are shown in Figure 1. We can see that selected prognostic genes from DNA methylation data are more stable in breast invasive carcinoma (BRCA) and uterine corpus endometrial carcinoma (UCEC). Furthermore, the Mann–Whitney–Wilcoxon (MWW) test was applied to test the differences between the p-values in gene expression data and DNA methylation data. The p-values of the MWW test are 4.28e-06 and 3.78e-04 in BRCA and UCEC, respectively. However, in skin cutaneous melanoma (SKCM), the stability between gene expression data and DNA methylation data was not significantly difference (MWW test *p*-value = 0.3). In conclusion, for the prognosis of cancer, DNA methylation data are more stable than gene expression data. Therefore, the DNA methylation data may be more suitable for a cancer prognostic study.

### 3.2. Module Networks of Three Cancers

The gene module network could uncover the cross-talks among the modules. In this work, we first used a rank-based method to construct a gene co-methylation network based on the DNA methylation data of cancer patients. Next, MCODE was used to detect the gene modules in the co-methylation network. Then, a permutation test was applied to calculate cross-talk among the gene modules. The module networks for the BRCA, SKCM, and UCEC are shown as follows. 

#### 3.2.1. Module Network of Breast Invasive Carcinoma

We constructed the gene co-methylation network by using the DNA methylation data of BRCA patients in TCGA. In this network, there are 16,850 nodes and 67,400 edges. The nodes’ degrees fit well with the power-law distributions with a correlation of 0.969 and R-square of 0.939 (Appendix A). Based on the co-methylation network, the gene module network of BRCA was constructed with 130 edges among the 97 modules (Appendix A).

#### 3.2.2. Module Network of Skin Cutaneous Melanoma

For SKCM, 68,124 co-methylation pairs among 17,031 genes were obtained. The power-law fit of nodes’ degrees with the number of nodes showed that the network was scale-free with a correlation of 0.975 and R-square of 0.948 (Appendix A). The module network of SKCM contained 110 edges among 97 modules (Appendix A).

#### 3.2.3. Module Network of Uterine Corpus Endometrial Carcinoma

The DNA methylation profiles of UCEC patients from TCGA were used to construct the gene co-methylation network. In the co-methylation network of UCEC, there were 17,081 nodes and 68,324 pairs. The co-methylation network of UCEC was also scale-free with a correlation of 0.973 and R-squared value of 0.946 (Appendix A). Based on the co-methylation network, the gene module network of UCEC was also constructed. There were 104 modules and 147 edges in the network (Appendix A).

### 3.3. The Core Gene Modules of Three Cancers

For the module network of each cancer, the K-shell algorithm was applied to identify the core gene modules. The K-shell method was shown to outperform other known centrality methods, including degree, betweenness, and PageRank in network-based analyses [31]. In the BRCA module network, there were 2 core gene modules that contained 46 genes (Appendix A). For SKCM, we obtained 4 core gene modules with 98 genes (Appendix A). In the module network of UCEC, there were 2 core gene modules, including 86 genes (Appendix A). These core gene modules of each cancer would be used as features for prognosis and molecular typing analysis.

### 3.4. Survival Analysis of Three Cancers

To evaluate the core gene modules, survival analysis of cancer datasets was performed for the three cancer types. In the gene module network of BRCA, there were 2 core gene modules. Based on the DNA methylation data of the 2 modules’ statistical values, the prognostic risks of cancer samples could be calculated. In the test dataset (390 patients) of BRCA, the hazard ratio (HR) of the High-risk group and Low-risk group divided by our method was 1.63, and the log rank p-value was 0.034 (Figure 2a). Based on the 4 core gene modules’ statistical values of the SKCM test dataset (234 patients), the HR of the two groups was 1.75, and the *p*-value of log rank was 4.2e-04 (Figure 2b). For UCEC, the HR of the High-risk group and Low-risk group was 2.53 with a log rank p-value of 0.0043 (Figure 2c). These results indicated that the core gene modules could distinguish the prognostic risks of cancer patients in three cancers. The good performance of these core gene modules also verified that our approach represented an improvement on previously reported methods.

### 3.5. Molecular Typing Results of Three Cancers

#### 3.5.1. Molecular Typing of Breast Invasive Carcinoma

In the clinic, breast cancer subtypes were defined by immunohistochemical detection of estrogen receptor (ER), progesterone receptor (PR), and human epidermal growth factor receptor 2 (HER2) expression [35]. The subtypes were luminal A (ER+, PR+/-, HER2-), luminal B (ER+, PR+/-, HER2+), HER2+ (ER-, PR-, HER2+), and basal-like (ER-, PR-, HER2-), respectively. According to the rule, 413 patients in the TCGA could be classified, including 249 patients with luminal A, 63 patients with luminal B, 17 patients with HRE2+, and 84 patients with basal-like disease. We first explored the relationship between this widely accepted molecular typing and prognosis of patients. The survival curve is shown in Figure 3, and the log rank *p*-value was 0.02, which indicated that this molecular typing was related to the prognosis of breast cancer patients. The silhouette coefficients of clustering under different k values were shown in Appendix A. In order to compared with this current typing criteria of BRCA, we used the core gene modules of BRCA as typing features and stratified breast cancer patients into 4 categories by the K-means algorithm. There were 112 patients in cluster 1, 134 patients in cluster 2, 150 patients in cluster 3, and 17 patients in cluster 4. The clustering result obtained by multidimensional scaling (MDS) is shown in Appendix A. The survival curve is shown in Figure 4, with the log rank p-value of 1e-04. This result indicated that there was a significant correlation between the classification of our method and the prognosis of patients. Compared with the current typing criteria, the molecular typing obtained by our method is more relevant to the prognosis of patients, demonstrating the reliability of the typing method proposed in this study.

#### 3.5.2. Molecular Typing of Skin Cutaneous Melanoma

The 4 core gene modules obtained in the SKCM module network were classified as features, and the silhouette coefficients of clustering under different k values were shown in Appendix A. When k is equal to 2, the silhouette coefficient is the highest (0.46). In other words, the classification effectiveness is optimal. The clustering result obtained by multidimensional scaling (MDS) is shown in Appendix A. There were 273 patients in cluster 1 and 195 patients in cluster 2. The survival curve of this molecular typing and patient prognosis is shown in Figure 5. The log rank p-value was 5e-04, proving the effectiveness of this method.

#### 3.5.3. Molecular Typing of Uterine Corpus Endometrial Carcinoma

According to the core gene modules obtained from the UCEC gene module network, we used the K-means algorithm for clustering. The silhouette coefficients of clustering under different k values are shown in Appendix A. When k is equal to 2 and 3, the silhouette coefficients are relatively high (0.58 and 0.56, respectively). We analyzed the clustering effectiveness in the two cases. The clustering results obtained by multidimensional scaling (MDS) are shown in Appendix A. When the samples were categorized into 2 subtypes (307 patients in cluster 1 and 121 patients in cluster 2), the log rank p-value was 0.2 (Appendix A). When the patients of UCEC were grouped into 3 subtypes (91 patients in cluster 1, 181 patients in cluster 2 and 156 patients in cluster 3), the survival analysis is shown in Figure 6 with a log rank p-value of 0.01. These results showed that it is more reasonable to categorize patients into 3 subtypes in UCEC.

### 3.6. Biological Functions of the Core Module Networks

We constructed cancer gene module networks based on DNA methylation data and then identified core gene modules using the K-shell method. These selected gene modules were applied to calculate the prognosis risk of patients, and survival analysis showed that our modules could significantly distinguish the prognosis risks of patients in three cancers. In addition, we used these core gene modules as typing features, and there was a significant correlation between the molecular types we identified and the prognosis of cancer samples. All of these results proved the reliability of our method. Then, the biological functions of genes contained in these modules will be explored next.

#### 3.6.1. Core Module Network Analysis of Breast Invasive Carcinoma

Using the K-shell approach, 2 core gene modules (Module 68 and Module 118) were obtained in the module network of BRCA. A total of 46 genes were included, as shown in Figure 7a. Ten of these genes have been reported to be involved in the development, metastasis and prognosis of breast cancer, as shown in Table 1.

In the core module network of BRCA, the gene *RCHY1* was studies. Because *RCHY1* is located at the junction of the two core modules, it is highly integrated in the core module network and connected to the gene *KDM1B*. The gene *KDM1B* is a histone demethylase that regulates histone lysine methylation and is an epigenetic marker that regulates gene expression and chromosomal function [36]. Studies have shown that *KDM1B* plays an important role in regulating DNA methylation and gene silencing in breast cancer [37]. Moreover, the *RCHY1* gene itself is a known oncogene, and this gene is a p53-induced ubiquitin–protein ligase that promotes p53 degradation. Loss of p53 function can directly lead to the development of malignant tumors [38]. However, the role of *RCHY1* in breast cancer has not been reported. We thought that the RCHY1 methylation degree is related to the prognosis of breast cancer; therefore, we divided the patients of BRCA into two groups according to the degree of methylation of the gene *RCHY1* (lower-methylation group and higher-methylation group). The survival analysis is shown in Figure 7b. The degree of methylation of the *RCHY1* gene is significantly correlated with the prognosis of breast cancer patients (cox p-value = 0.0098), and the prognosis of patients with a higher degree of methylation of *RCHY1* is better than that of patients with lower methylation of *RCHY1*. To some extent, this result explained the carcinogenic mechanism of the gene *RCHY1*. In other words, the hypomethylation of the *RCHY1* promoter region leads to its overexpression in cancer development, which promotes the binding of *RCHY1* and p53 to the degradation of p53 protein and then the malignant development of the tumor. The expression of the gene *RCHY1* in higher-methylation group and lower-methylation group was shown in Figure 7c. It can be clearly seen that the *RCHY1* expression of patients in lower-methylation group is significantly higher than that in the higher-methylation group (MWW test *p*-value = 6.5e-03). This conclusion could also prove the validity of the results of our study.

#### 3.6.2. Core Module Network Analysis of Skin Cutaneous Melanoma

We identified 4 core gene modules in the module network of SKCM: Module 73, Module 98, Module 122, and Module 123, including 96 genes, as shown in Figure 8. Ten of these genes are associated with metastasis and prognosis of SKCM (Table 2).

Similar to Section 3.6.1, we recognized 4 potential prognostic genes in the core module network of SKCM, namely, *APPL1*, *BCL2L2*, *TRIM3,* and *UEVLD*. The 4 genes are located at junction among modules with higher degrees, and they are linked to known genes that are associated with the prognosis of SKCM (*RHOJ*, *PROX1*, *TBK1*, and *UGDH*), as shown in Figure 8. We hypothesized that the methylation levels of these 4 genes could be used as biomarkers to predict the prognosis of patients with SKCM. Survival analysis is shown in Figure 9. The methylation degrees of *APPL1*, *BCL2L2*, *TRIM3,* and *UEVLD* were significantly correlated with the prognosis of SKCM patients. The p-values of cox regression were 0.001, 5e-05, 7e-04, and 3e-05, respectively, and patients with a higher methylation degree of these genes had a better prognosis than those with a lower degree of methylation.

#### 3.6.3. Core Module Network Analysis of Uterine Corpus Endometrial Carcinoma

The 2 core gene modules were identified in the UCEC module network by the K-shell method. The modules were Module 37 and Module 68, as shown in Figure 10a, which included 86 genes. Among these genes, 5 genes were known to be associated with the development and prognosis of UCEC, which are shown in Table 3.

In the core module network of UCEC, the gene *POP1* was located at the junction of the two modules and has the highest degrees in the core module network. We used the methylation degree of POP1 as a prognostic biomarker, and the survival analysis is shown in Figure 10b. The p-value of Cox regression was 0.016, and the prognosis of patients with a lower degree of methylation was better than that of patients with higher methylation. Furthermore, in Module 68, although the gene *PSMB9* was not located at the junction of the two modules, it was linked to three genes known to be associated with the development and prognosis of UCEC (*AURKA*, *FBXW7,* and *JAG1*). The survival analysis is shown in Figure 10c. The methylation degree of PSMB9 is significantly correlated with the prognosis of UCEC patients (Cox *p*-value = 0.011), and the prognosis of patients with a lower degree of methylation is better than that of patients with higher methylation.

## 4. Conclusions

The importance of exact prognosis and rational molecular typing for the treatment of cancer are self-evident. Because of the heterogeneity of cancer, it is impossible to predict the prognosis and classification of patients if only some clinical indicators are relied on. Some gene signatures associated with cancer prognosis and classification were identified based on high-throughput data, but these signatures selected from one dataset are not applicable in other datasets [5], and most of these genes are not oncogenes but noise signals [6]. Studies have shown that gene modules are more stable than gene signatures [13], and cross-talk exists between two modules. However, few studies have noted this phenomenon. Currently, the abnormality of DNA methylation has been shown to be associated with cancer prognosis [9,10], and the accumulation of DNA methylation data provides an opportunity to study cancer at the epigenetic level.

The study was the first to systematically evaluate the stabilities of DNA methylation data and gene expression data in BRCA, SKCM, and UCEC and proved that DNA methylation data may be more stable in cancer prognosis. Then, the DNA methylation data were used to construct gene co-methylation networks for the three cancers, and the gene modules were identified in three co-methylation networks. Next, the permutation test was used to calculate the cross-talk between every two modules; therefore, module networks were forming. Then, we found the core gene modules in the module network by the K-shell method, and these core gene modules are used as features to study the prognosis and molecular typing of cancer. Finally, we found 2 core gene modules in BRCA, 4 core gene modules in SKCM, and 2 core gene modules in UCEC. These core modules can significantly distinguish patients’ prognoses. Then, these core modules as clustering features were used to classify three cancers by the K-means algorithm. The typing results were also significantly correlated with the prognosis of cancer patients. In addition, after analyzing the topology of the core module networks in three cancers, we identified DNA methylation prognostic biomarkers in three cancers. These results demonstrate the effectiveness of our method in cancer prognosis and molecular typing.

Not surprisingly, our study also has certain flaws. The main problem is that no suitable independent datasets have been found. Although abundant DNA methylation cancer data are available, there are few datasets with prognostic information of cancer patients. Therefore, we did not find suitable independent datasets for the three cancers to verify.

## Figures and Tables

**Figure 1 genes-10-00571-f001:**
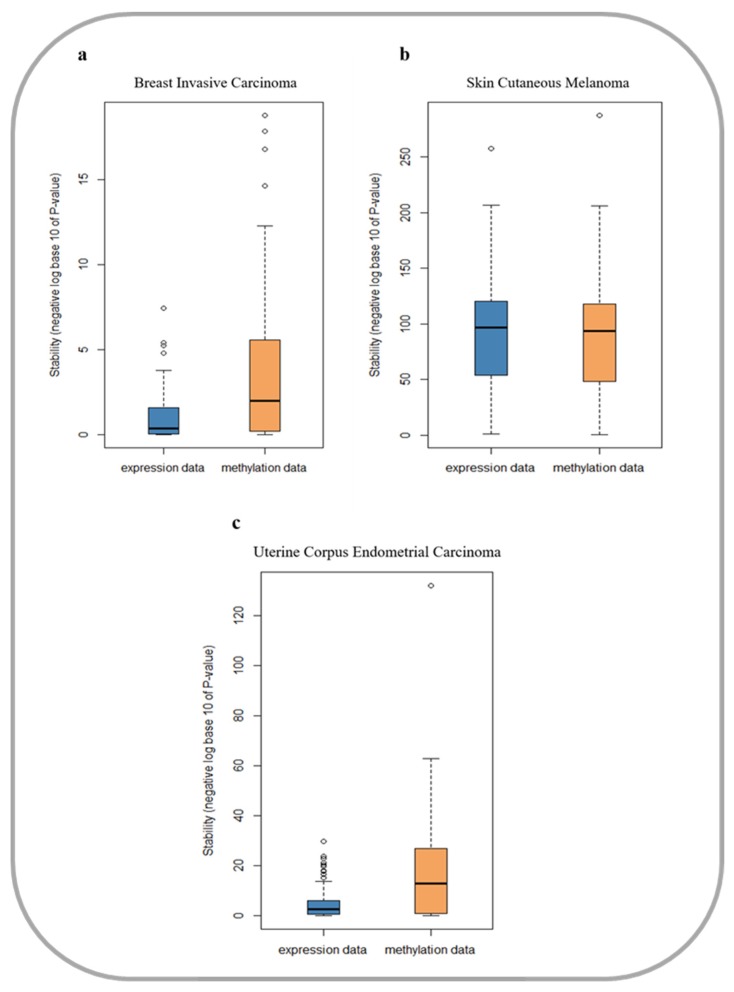
Stability verification in prognosis of gene expression data and methylation data in three cancer datasets. (**a**) Evaluation results in breast invasive carcinoma dataset; (**b**) Evaluation results in skin cutaneous melanoma dataset; (**c**) Evaluation results in uterine corpus endometrial carcinoma dataset.

**Figure 2 genes-10-00571-f002:**
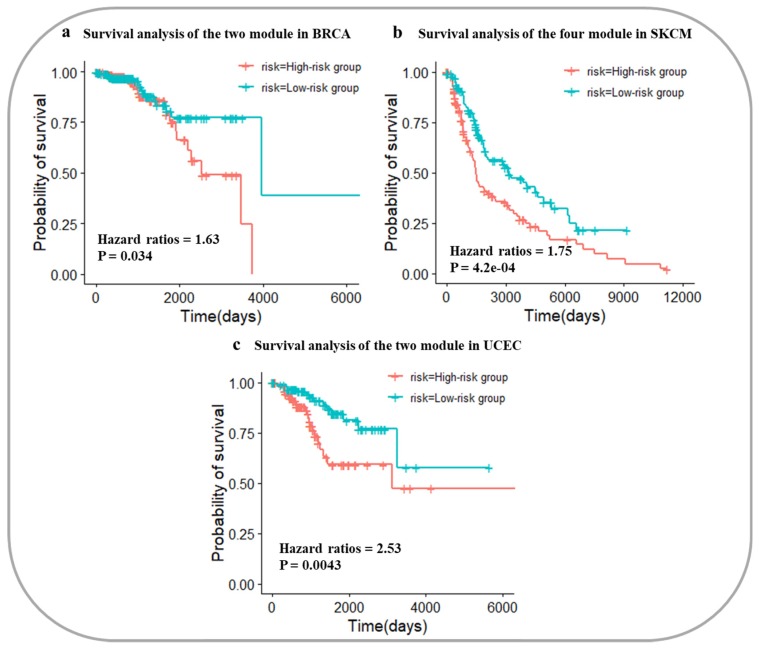
Survival analysis in the three cancer datasets based on core gene modules. (**a**) Survival analysis in the breast invasive carcinoma dataset; (**b**) Survival analysis in the skin cutaneous melanoma dataset; (**c**) Survival analysis in the uterine corpus endometrial carcinoma dataset. BRCA: breast invasive carcinoma; SKCM: skin cutaneous melanoma; UCEC: uterine corpus endometrial carcinoma.

**Figure 3 genes-10-00571-f003:**
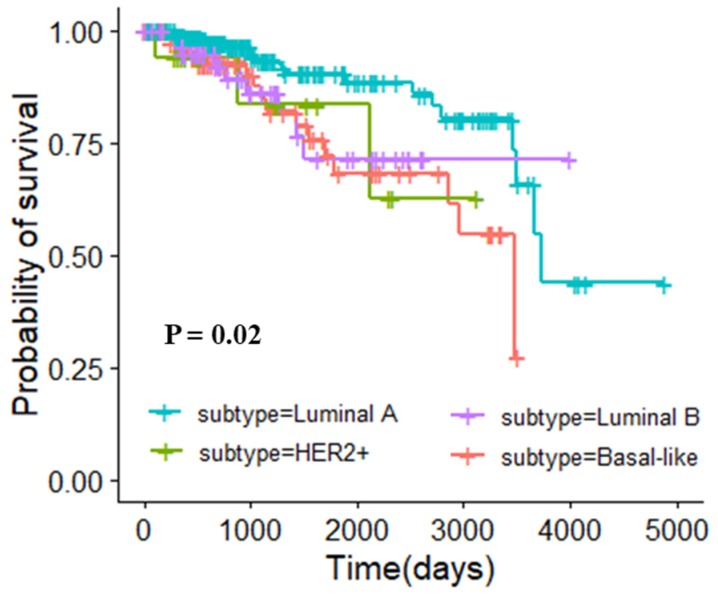
Survival analysis of primitive typing in breast invasive carcinoma dataset.

**Figure 4 genes-10-00571-f004:**
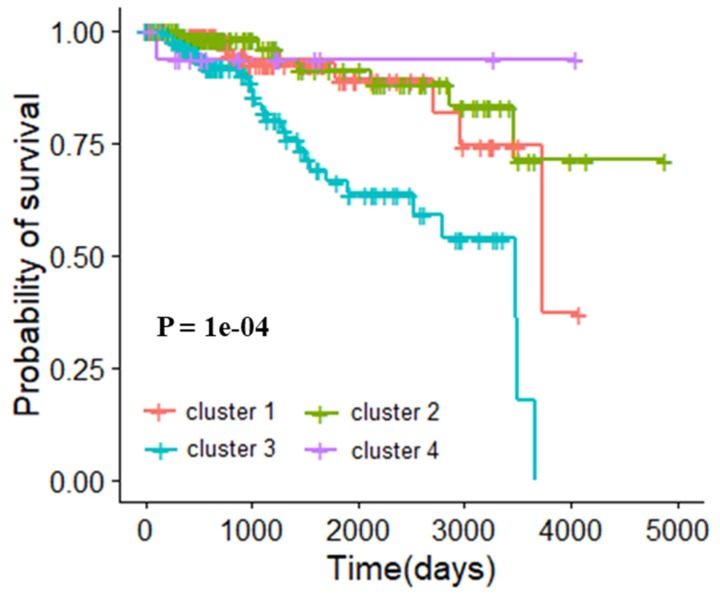
Survival analysis of classification by the K-means clustering algorithm in breast invasive carcinoma dataset.

**Figure 5 genes-10-00571-f005:**
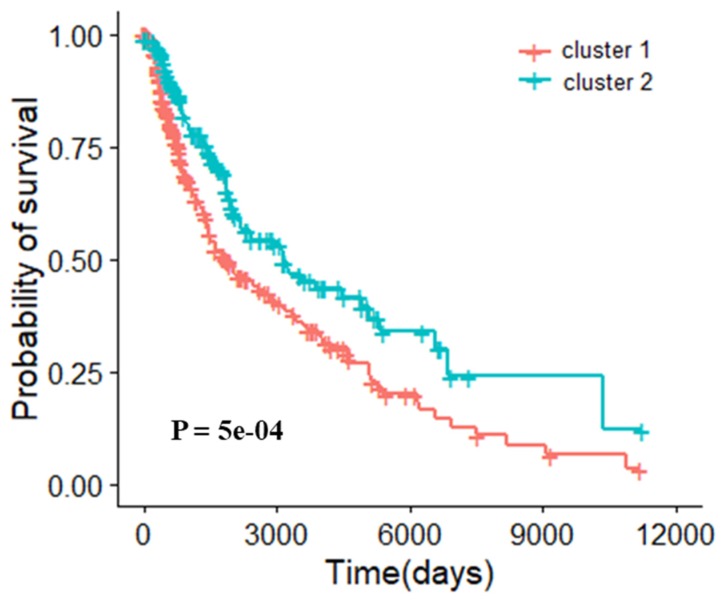
Survival analysis of different types of skin cutaneous melanoma samples after clustering (K = 2).

**Figure 6 genes-10-00571-f006:**
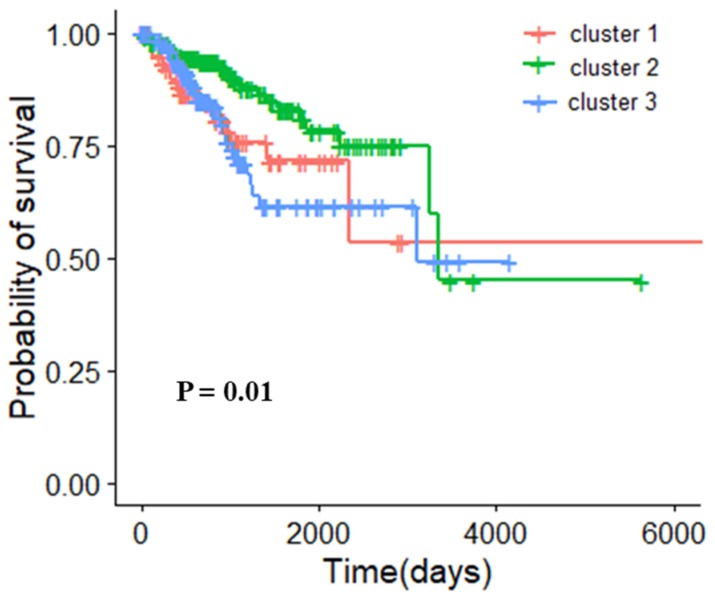
Survival analysis of different types of uterine corpus endometrial carcinoma samples after clustering (K = 3).

**Figure 7 genes-10-00571-f007:**
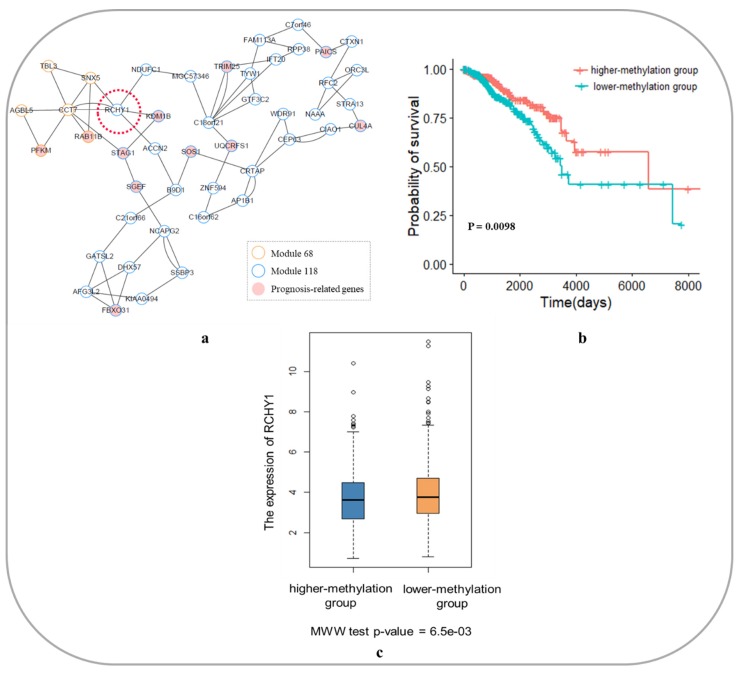
Core module network and survival analysis of *RCHY1* as a biomarker in breast invasive carcinoma. (**a**) Core module network of breast invasive carcinoma; (**b**) Survival analysis of *RCHY1* as a biomarker in breast invasive carcinoma; (**c**) The expression of the gene *RCHY1* in higher-methylation group and lower-methylation group. MWW: Mann–Whitney–Wilcoxon.

**Figure 8 genes-10-00571-f008:**
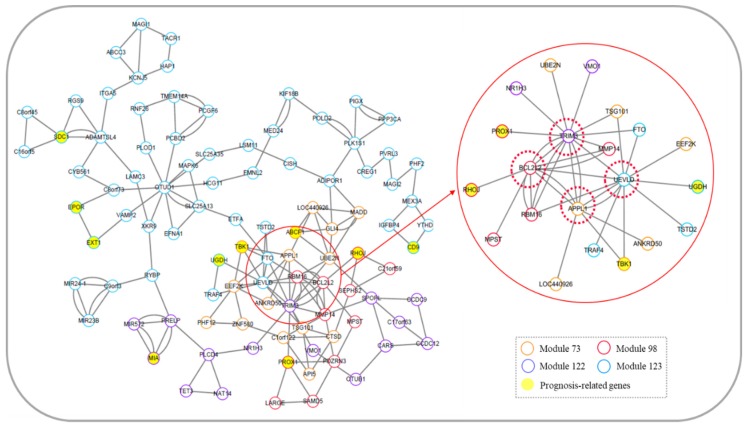
Core module network of skin cutaneous melanoma. The red solid circle indicates the local network topology diagram associated with genes *APPL1*, *BCL2L2*, *TRIM3*, and *UEVLD*.

**Figure 9 genes-10-00571-f009:**
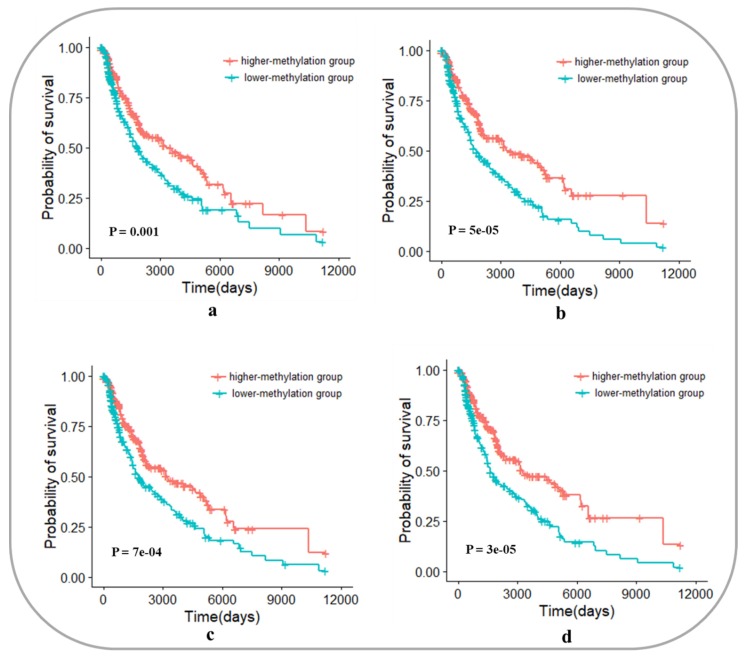
Survival analysis of *APPL1*, *BCL2L2*, *TRIM3*, and *UEVLD* as biomarkers in skin cutaneous melanoma. (**a**) Survival analysis of *APPL1* as a biomarker in skin cutaneous melanoma; (**b**) Survival analysis of *BCL2L2* as a biomarker in skin cutaneous melanoma; (**c**) Survival analysis of *TRIM3* as a biomarker in skin cutaneous melanoma; (**d**) Survival analysis of *UEVLD* as a biomarker in skin cutaneous melanoma.

**Figure 10 genes-10-00571-f010:**
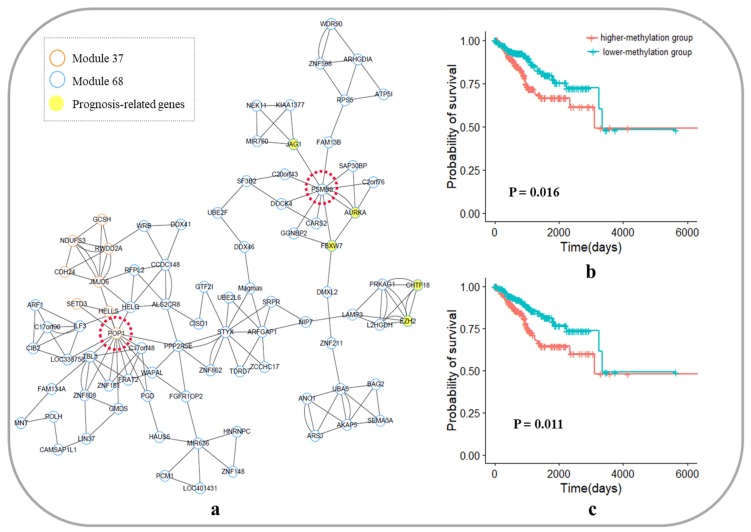
Core module network and survival analysis of *POP1* and *PSMB9* as biomarkers in uterine corpus endometrial carcinoma. (**a**) Core module network of uterine corpus endometrial carcinoma; (**b**) Survival analysis of *POP1* as a biomarker in uterine corpus endometrial carcinoma; (**c**) Survival analysis of *PSMB9* as a biomarker in uterine corpus endometrial carcinoma.

**Table 1 genes-10-00571-t001:** Known genes associated with breast invasive carcinoma development, metastasis, and prognosis in the core module network.

Gene Symbol	Full Name	Gene Function	References
*CUL4A*	cullin 4A	CUL4A is the ubiquitin ligase component of a multimeric complex involved in the degradation of DNA damage-response proteins. Overexpression of CUL4A is associated with poor prognosis in BRCA.	[39,40,41]
*FBXO31*	F-box protein 31	This gene is a member of the F-box family. The FBXO31 dysregulation is associated with the development of BRCA and is a candidate tumor suppressor gene.	[42]
*KDM1B*	lysine demethylase 1B	This gene is a flavin-dependent histone demethylase that regulates histone lysine methylation. Abnormal DNA methylation of KDM1B is associated with poor prognosis in BRCA.	[37,43]
*PAICS*	phosphoribosylaminoimidazole carboxylase and phosphoribosylaminoimidazolesuccinocarboxamide synthase	PAICS is an enzyme required for biosynthesis of purine. It is associated with poor prognosis of BRCA.	[44,45]
*PFKM*	phosphofructokinase, muscle	The gene is associated with increased risk of BRCA.	[46]
*RAB11B*	RAB11B, member RAS oncogene family	RAB11B is a member of the Ras superfamily of small GTP-binding proteins. It is associated with metastasis of breast cancer.	[47,48]
*SGEF*	Rho guanine nucleotide exchange factor 26	The gene encodes a member of the Rho-guanine nucleotide exchange factor family and is associated with cancer invasion.	[49,50]
*SOS1*	SOS Ras/Rac guanine nucleotide exchange factor 1	The protein encoded by gene SOS1 is a guanine nucleotide exchange factor of RAS protein, a membrane protein that binds to guanine nucleotides and participates in signal transduction pathways. It is associated with invasion and metastasis of breast cancer.	[51,52]
*STAG1*	stromal antigen 1	This gene is a member of the SCC3 family and is expressed in the nucleus. Its overexpression might be regarded as a tumor marker in BRCA.	[53]
*TRIM25*	tripartite motif containing 25	Expression of the TRIM25 is upregulated in response to estrogen, and the TRIM25 is as a diver of poor outcome in BRCA.	[54,55]
*UQCRFS1*	ubiquinol-cytochrome c reductase, Rieske iron-sulfur polypeptide 1	The gene is a key subunit of the cytochrome bc1 complex and is associated with the development of breast cancer.	[56]

**Table 2 genes-10-00571-t002:** Known genes associated with skin cutaneous melanoma development, metastasis, and prognosis in the core module network.

Gene Symbol	Full Name	Gene Function	References
*ABCF1*	ATP binding cassette subfamily F member 1	The protein encoded by this gene is a member of the ATP-binding cassette (ABC) transporters superfamily, and it is involved in the development of SKCM.	[57]
*CD9*	CD9 molecule	This gene encodes a transmembrane 4 superfamily member whose expression plays a key role in inhibiting the metastasis of SKCM.	[58,59]
*EPOR*	erythropoietin receptor	The gene encodes an erythropoietin receptor and is a member of the cytokine receptor family. Its dysregulation is associated with metastasis and prognosis of cutaneous melanoma.	[60,61]
*EXT1*	exostosin glycosyltransferase 1	This gene encodes an endoplasmic reticulum-resident type II transmembrane glycosyltransferase. It is associated with the development of SKCM.	[62]
*MIA*	MIA SH3 domain containing	It is a cartilage-derived retinoic acid-sensitive protein. This gene is associated with metastasis of cutaneous melanoma and is highly sensitive tumor markers for monitoring of patients with SKCM.	[63,64]
*PROX1*	prospero homeobox 1	The protein encoded by this gene is a member of the homeobox transcription factor family. It is related to the prognosis of SKCM.	[65,66]
*RHOJ*	ras homolog family member J	This gene encodes a Rho family GTP-binding protein that is involved in the invasion and metastasis of cutaneous melanoma.	[67,68]
*SDC1*	syndecan 1	The protein encoded by this gene is a transmembrane (type I) heparan sulfate proteoglycan, and it is associated with invasion of SKCM.	[69,70]
*TBK1*	TANK binding kinase 1	The gene is a member of the atypical IκB kinase family and is associated with invasion and migration of SKCM.	[71]
*UGDH*	UDP-glucose 6-dehydrogenase	The protein encoded by this gene converts UDP-glucose into UDP-glucuronic acid, thereby participating in the biosynthesis of glycosaminoglycans. It is related to the development of SKCM.	[72]

**Table 3 genes-10-00571-t003:** Known genes associated with uterine corpus endometrial carcinoma development, metastasis, and prognosis in the core module network.

Gene Symbol	Full Name	Gene Function	References
*AURKA*	aurora kinase A	The protein encoded by this gene is a cell cycle-regulated kinase. It has a significant relationship with the prognosis of UCEC and can be used as a clinical biomarker for UCEC.	[73,74]
*CHTF18*	chromosome transmission fidelity factor 18	The protein encoded by the gene is a component of a replication factor C (RFC) complex. The mutation of this gene is associated with the pathogenesis of UCEC.	[75]
*EZH2*	enhancer of zeste 2 polycomb repressive complex 2 subunit	This gene encodes a member of the Polycomb-group (PcG) family and is related to the prognosis of UCEC. It can be used as a prognostic marker for UCEC.	[76,77]
*FBXW7*	F-box and WD repeat domain containing 7	This gene encodes a member of the F-box protein family. It is associated with the development and prognosis of UCEC.	[78,79]
*JAG1*	jagged canonical Notch ligand 1	This gene encodes a jagged 1 protein. It is closely related to the invasion and prognosis of UCEC.	[80]

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
