# Peer review of "DNA Methylation Module Network-Based Prognosis and Molecular Typing of Cancer"

_genes, 2019, doi:10.3390/genes10080571_

Reviewer 1 Report

The paper entitled “DNA methylation module network-based prognosis and molecular typing of cancer” is focus on novel approach to the comparison of the stability of DNA methylation data and gene expression data. The Authors in general indicate the weaknesses of the applications/methods used for the high- throughput sequencing data analysis and present their own method to examine the stability of gene expression data and DNA methylation data. The novel methodological approach to data analysis has been validated on three cancer types: (i) breast invasive carcinoma (BRCA), (ii) skin cutaneous melanoma (SKCM) and (iii) uterine corpus endometrial carcinoma (UCEC).

Although the data obtained after analysis are very interesting and underline the importance of using DNA methylation data for cancer prognosis development and selection of patients to the specific therapies it is not clear why the precious approach is improper. The analysis done by the other approach are insufficiently presented thus the comparison between the methods is not possible.

In my opinion the content of the manuscript suggests submission of the article to the journal with stronger bioinformatic profile than Genes as the main aim of the paper is presentation of novel analysis approach and estimation of biostatistic tools used for analysis is necessary.

Reviewer 2 Report

The manuscript "DNA Methylation module network-based prognosis and molecular typing of cancer" by Cui et al., presents the results of gene expression and DNA methylation data on three different types of cancer patients. The authors show that promoter DNA methylation can offer a more stable tool for cancer prognosis respect to gene expression. In addition, the authors created a methylome based gene network which lead to the identification of gene modules containing important biomarkers for the prognosis of breast invasive carcinoma (BRCA), skin cutaneous melanoma (SKCM) and uterine corpus endometrial carcinoma (UCEC). The design of the analysis and the presented methods are sound and statistically supported.

Despite the good quality of the presented results there are some minor points that should be addressed before publication as listed below:

1) It appears that the authors used the DNA methylation data without considering the expression outcome of methylation. We should expect an anticorrelation between DNA promoter methylation and gene expression, so it would have made sense to consider only the genes which showed an opposite trend between DNA methylation and expression. Did the authors consider this point and how the results would change considering this filter?

2) In figure 4 the authors show that the BRCA samples could be divided in 4 clusters using a K-means approach. Is there any relationship between those clusters and the basal/luminal-A/luminal-B/HER2+ groups?

3) How did the authors choose K=4 for the clusterization of the BRCA dataset? The silhouette plot which is described for the other cancer types is not showed for BRCA. The authors should add it to the results and comment it.

4) At lines 300-302, the authors write "In other words, the hypomethylation of the RCHY1 promoter region leads to its overexpression in cancer development" however they did not show the expression profile of RCHY1 in their dataset which would confirm their claim.

5) There is no reference in the methods about the programs/algorithms used to create the methylome network nor the statistical analyses. The authors should mention them and cite the corresponding references.

6) The discussions are lacking a more detailed comparison with already published work on the use of methylation data for biomarkers detection in cancer. For example just for BRCA I could easily find two papers (https://www.ncbi.nlm.nih.gov/pmc/articles/PMC6026876/, https://bmccancer.biomedcentral.com/articles/10.1186/s12885-019-5403-0) that are related to the present work which might be worth to discuss. The authors should improve the discussions by looking also for other work in BRCA, SKCM and UCEC.

Additional suggestions (optional) that might be interesting for the readers:

1) The authors show that using methylation data it is possible to find new and stable biomarkers for prognosis and that the clusters generated with the methylation data are able to differentiate high risk vs low risk groups on BRCA, SKCM and UCEC. It would be interesting to test if combining expression and methylation data for the those genes would further improve the results. I.e. is the combination of methylation and expression data giving more power for prognosis?

Author Response

Round  2

Reviewer 1 Report

I recommend publication of the paper in the current form.